# High-performing neural network models of visual cortex benefit from high latent dimensionality

**Eric Elmoznino**  **\*, Michael F. Bonner**

Department of Cognitive Science, Johns Hopkins University, Baltimore, Maryland, United States of America

\* eric.elmoznino@gmail.com

## Abstract

Geometric descriptions of deep neural networks (DNNs) have the potential to uncover core representational principles of computational models in neuroscience. Here we examined the geometry of DNN models of visual cortex by quantifying the latent dimensionality of their natural image representations. A popular view holds that optimal DNNs compress their representations onto low-dimensional subspaces to achieve invariance and robustness, which suggests that better models of visual cortex should have lower dimensional geometries. Surprisingly, we found a strong trend in the opposite direction—neural networks with high-dimensional image subspaces tended to have better generalization performance when predicting cortical responses to held-out stimuli in both monkey electrophysiology and human fMRI data. Moreover, we found that high dimensionality was associated with better performance when learning new categories of stimuli, suggesting that higher dimensional representations are better suited to generalize beyond their training domains. These findings suggest a general principle whereby high-dimensional geometry confers computational benefits to DNN models of visual cortex.

## Author summary

The effective dimensionality of neural population codes in both brains and artificial neural networks can be far smaller than the number of neurons in the population. In vision, it has been argued that there are crucial benefits of representing images using codes that are as simple and low-dimensional as possible, allowing representations to emphasize key semantic content and attenuate irrelevant perceptual details. However, there are competing benefits of high-dimensional codes, which can capture rich perceptual information to better support an open-ended set of visual behaviors. We quantified the effective dimensionality of neural networks created using a diverse set of training paradigms and compared these networks with image-evoked codes in the visual cortex of both monkeys and humans. Our findings revealed striking benefits of networks with higher effective dimensionality, which were consistently better predictors of cortical activity patterns and were able to readily learn new categories of images outside of their training domains. Together, these findings demonstrate the computational benefits of high-dimensional sensory codes, and they suggest that current neural network models of visual cortex may

**Data Availability Statement:** The neural electrophysiology datasets are publicly available and were collected in prior work by Majaj et al and Freeman et al. doi:10.1523/JNEUROSCI.5181-14.2015 and DOI: 10.1038/nn.3402. It is imported

automatically using the BrainScore version 0.2 Python library through code in our publicly available project repository. The fMRI data was collected in prior work by Bonner et al and is publicly available here: https://osf.io/ug5zd/. All code used for this project has been made publicly available on GitHub: https://github.com/EricElmoznino/encoder_dimensionality.

**Funding:** The author(s) received no specific funding for this work.

**Competing interests:** The authors have declared that no competing interests exist.

be better understood in terms of the richness of their representations rather than the details of their training tasks.

## Introduction

Deep neural networks (DNNs) are the predominant framework for computational modeling in neuroscience [1–5]. When using DNNs to model neural systems, one of the fundamental questions that researchers hope to answer is: What core factors explain why some DNNs succeed and others fail? Researchers often attribute the success of DNNs to explicit design choices in a model's construction, such as its architecture, learning objective, and training data [2, 4–14]. However, an alternative perspective explains DNNs through the geometry of their latent representational subspaces, which abstracts over the details of how networks are constructed [15–19]. Here we sought to understand the geometric principles that underlie the performance of DNN models of visual cortex.

We examined the geometry of DNNs by quantifying the dimensionality of their representational subspaces. DNN models of vision contain thousands of artificial neurons, but their representations are known to be constrained to lower-dimensional subspaces that are embedded within the ambient space of the neural population [20, e.g.]. Many have argued that DNNs benefit from representing stimuli in subspaces that are as low-dimensional as possible, and it is proposed that low dimensionality improves a network's generalization performance, its robustness to noise, and its ability to separate stimuli into meaningful categories [20–32]. Similar arguments have been made for the benefits of low-dimensional subspaces in the sensory, motor, and cognitive systems of the brain [33–39]. However, contrary to this view, there are also potential benefits of high-dimensional subspaces, including the efficient utilization of a network's representational resources and increased expressivity, making for a greater number of potential linear readouts [40–45].

We wondered whether the dimensionality of representational subspaces might be relevant for understanding the relationship between DNNs and visual cortex and, if so, what level of dimensionality performs best. To answer these questions, we measured the latent dimensionality of DNNs trained on a variety of supervised and self-supervised tasks using multiple datasets, and we assessed their accuracy at predicting image-evoked activity patterns in visual cortex for held-out stimuli using both monkey electrophysiology and human fMRI data. We discovered a powerful relationship between dimensionality and accuracy: specifically, we found that DNNs with higher latent dimensionality explain more variance in the image representations of high-level visual cortex. This was true even when controlling for model depth and the number of parameters in each network, and it could not be explained by overfitting because our analyses explicitly tested each network's ability to generalize to held-out stimuli. Furthermore, we found that high dimensionality also conferred computational benefits when learning to classify new categories of stimuli, providing support for its adaptive role in visual behaviors. Together, these findings suggest that high-performing computational models of visual cortex are characterized by high-dimensional representational subspaces, allowing them to efficiently support a greater diversity of linear readouts for natural images.

## Results

### Dimensionality and alignment in computational brain models

We set out to answer two fundamental questions about the geometry of DNNs in computational neuroscience. First, is there a relationship between latent dimensionality and DNN

performance? Second, if latent dimensionality is indeed related to DNN performance, what level of dimensionality is better? In other words, do DNN models of neural systems primarily benefit from the robustness and invariance of low-dimensional codes or the expressivity of high-dimensional codes? To explore the theoretical issues underlying these questions, we first performed simulations that illustrate how the geometry of latent subspaces might influence the ability of representational models to account for variance in brain activity patterns.

For our simulations, we considered a scenario in which all brain representations and all relevant computational models are sampled from a large subspace of image representations called the *natural image subspace*. Here, we use the term subspace to describe the lower-dimensional subspace spanned by the major principal components of a system with higher ambient dimensionality (e.g., neurons). We sampled observations from this natural image subspace and projected these observations onto the dimensions spanned by two smaller subspaces called the *ecological subspace* and the *model subspace*. Projections onto the ecological subspace simulate image representations in the brain, and projections onto the model subspace simulate image representations in a computational model. We analyzed these simulated data using a standard approach in computational neuroscience, known as the encoding-model approach. Specifically, we mapped model representations to brain representations using cross-validated linear regression. This analysis yielded an encoding score, which is the explained variance for held-out data in the cross-validated regression procedure. Computational models with higher encoding scores have better performance when predicting brain representations for held-out data. Further details regarding the theoretical grounding and technical implementation of our simulations are provided in S2 & S3 Text.

Using this simulation framework, we can now illustrate how two important factors might be related to the performance of computational brain models: effective dimensionality and alignment pressure. *Effective dimensionality* (ED) is a continuous measurement of the number of principal components needed to explain most of the variance in a dataset, and it is a way of estimating *latent* dimensionality in our analyses (see Fig 1a). A model with low ED encodes a relatively small number of dimensions whose variance is larger than the variance attributed to noise (i.e., whose signal-to-noise ratio (SNR) is high). In contrast, a model with high ED encodes many dimensions with high SNR. *Alignment pressure* (AP) quantifies the probability that the high SNR dimensions from a pair of subspaces will be aligned, as depicted in Fig 1b. For example, if the AP between a model subspace and the ecological subspace is high, it means that the model is likely to encode the same dimensions of image representation as those observed in the brain.

Nearly all representational modeling efforts in computational neuroscience seek to optimize AP. For example, when researchers construct models through deep learning or by specifying computational algorithms, the hope is that the resulting model will encode representational dimensions that are strongly aligned with the representations of a targeted brain system. However, if one allows for linear transformations when mapping computational models to brain systems—a procedure that may, in fact, be necessary for evaluating such models [6]—then there are possible scenarios in which model performance can be primarily governed by ED.

To understand how ED can influence model performance, it is helpful to first consider two extreme cases. At one extreme, models with an ED of 1 can explain, at best, a single dimension of brain representation and can only do so when AP is extremely high. Such a model would need to encode a dimension that was *just right* to explain variance in a targeted brain system. At the other extreme, a model with very high ED could potentially explain many dimensions of brain representation and could do so with weaker demands on AP. This means that models

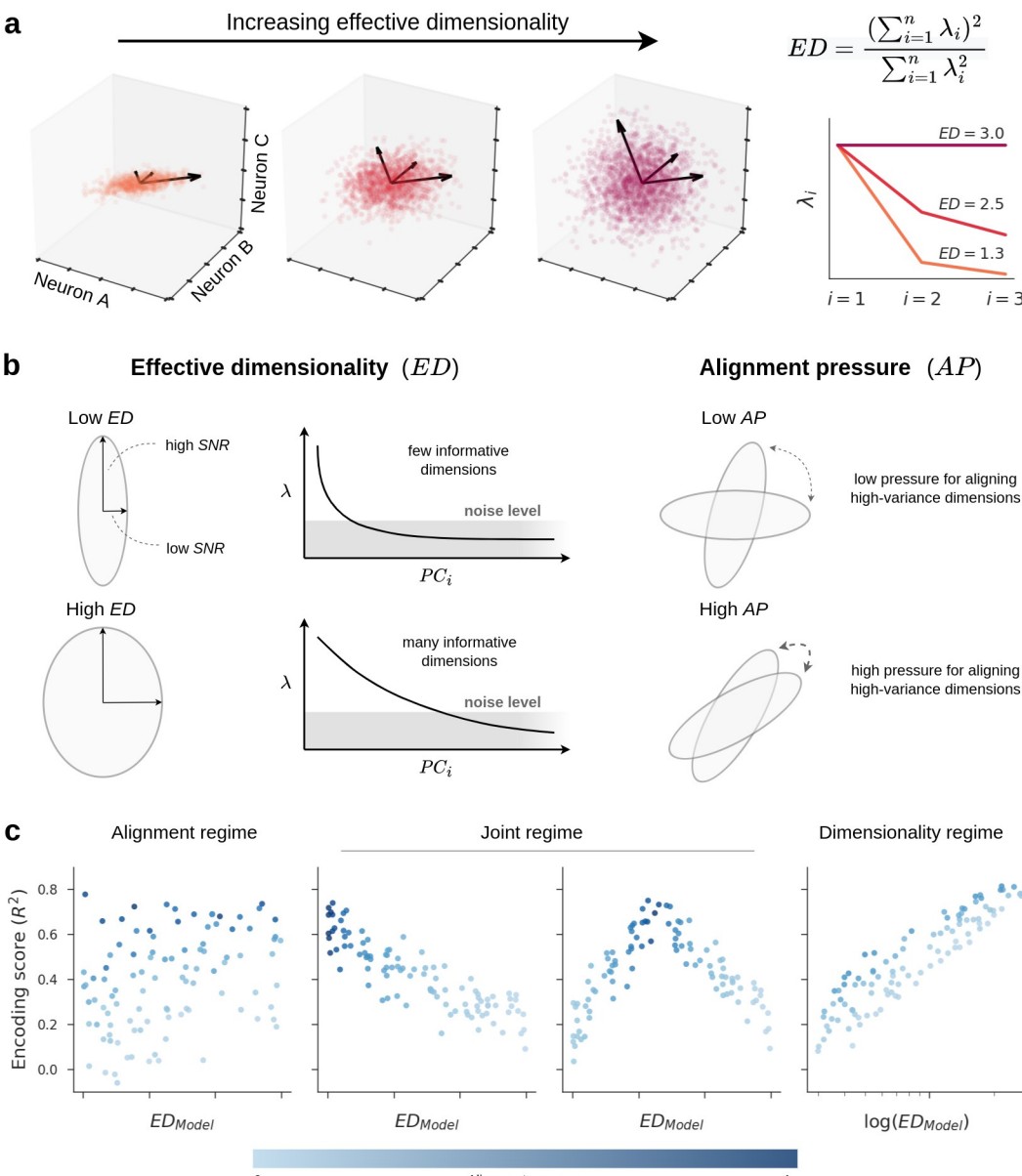

**Fig 1. A theory of latent dimensionality and encoding performance.** a: This panel illustrates effective dimensionality (ED) for a hypothetical population of three neurons. The data points correspond to stimuli, and the plot axes indicate the firing rates of neurons in response to these stimuli. The leftmost plot shows a scenario where the firing rates of the neurons are highly correlated and primarily extend along a single direction, resulting in an ED close to 1. The opposite scenario is shown in the rightmost plot where variance in neural responses is equally distributed across all directions, resulting in an ED of 3. On the right, we show the eigenspectra ($\lambda_i$) in each scenario and the equation that describes how ED is computed from these eigenspectra. b: Our simulations examine two geometric properties: effective dimensionality (ED) and alignment pressure (AP). ED is a summary statistic that indicates the number of features accurately encoded by an ecological or model subspace (i.e., it is a way of estimating latent dimensionality). The eigenspectrum of a low-dimensional subspace decays quickly, suggesting that most features are dominated by noise and, therefore, poorly encoded, whereas the eigenspectrum of a high-dimensional subspace has high variance spread along a large number of dimensions. AP determines the alignment of high-variance dimensions across two subspaces. Pairs of subspaces with low AP are sampled independently with little bias for their signal dimensions to align, whereas pairs of subspaces with high AP are more likely to have substantial overlapping variance along their signal dimensions. c: Depending on the distributions of ED and AP in empirical models, our simulations predict different outcomes for the relationship between model ED and encoding performance. In the Alignment regime, model performance is predominantly driven by the alignment of the meaningful, signal dimensions in the model and the brain, with little to no influence of latent dimensionality. Most modeling efforts in computational neuroscience implicitly assume that models operate in the Alignment regime. Another possibility is that models operate in a Joint regime,

in which there exists some optimal dimensionality at which model representations are more likely to be aligned with the brain (perhaps a low dimensionality as in the plot on the left, or an intermediate dimensionality as in the plot on the right). This is the implicit assumption behind efforts to explain brain representations with models that compress latent dimensionality (such as autoencoders). A third possibility, which has been largely overlooked, is that models operate in a Dimensionality regime, in which models with higher latent dimensionality are more likely to contain the same representational dimensions that were sampled in a neuroscience experiment. Note that the Dimensionality regime occurs when there is large variance in model ED, so we use a logarithmic scale on the x-axis for this regime.

with extremely high ED have a higher probability of performing well and need only be partially aligned with a targeted brain system.

The relative contributions of ED and AP will depend on their empirical distribution in actual computational models trained to predict real neural data. To better anticipate the possible outcomes, we varied our simulation parameters and identified distinct regimes for the relationship between ED, AP, and the performance of computational brain models (Fig 1c).

In the *Alignment regime*, the ED of computational models varies significantly less than their AP, such that AP predominantly drives performance in predicting neural activity. This perspective implicitly underlies most deep learning approaches for modeling visual cortex, which emphasize factors affecting the alignment between a model and the brain, such as architecture, layer depth, learning objective, and training images [2, 4–14, e.g.]. The alignment-based perspective does not entail any specific predictions about ED and, thus, suggests the null hypothesis that ED and encoding scores are unrelated (Fig 1c left panel).

Alternatively, models might inhabit a *Joint regime* where ED and AP are entangled, such that there exists some optimal dimensionality at which model representations are more likely to be aligned with the brain. Previous work has proposed that both biological and artificial vision systems gain computational benefits by representing stimuli in low-dimensional subspaces [15, 20, 37]. For instance, it has been hypothesized that dimensionality reduction along the visual hierarchy confers robustness to incidental image features [21–25, 30, 31]. This dimensionality-reduction hypothesis implicitly underlies a wide variety of machine learning methods that attempt to encode complex stimuli using a small set of highly informative dimensions (e.g., autoencoders) [27, 28, 32]. The strongest version of the low-dimensionality perspective predicts that ED and encoding scores will be negatively correlated or exhibit an inverted U-shape, since models with relatively low-dimensional subspaces would tend to be better aligned with the representations of visual cortex (Fig 1c middle panels).

A final possibility is that of a *Dimensionality regime*. This can occur if the computational models under consideration vary significantly in terms of ED and are sufficiently constrained to make the baseline probability of partially overlapping with visual cortex non-negligible (i.e., they have some moderate level of AP). In this case, ED will exert a strong, positive influence on expected encoding performance (Fig 1c right panel). It is unknown whether ED is a relevant factor for convolutional neural network (CNN) models of visual cortex, and, if so, whether high-dimensional representations lead to better or worse models. Our simulations suggest several possible outcomes depending on the empirical distribution of ED and AP, including a previously unexplored scenario where high latent dimensionality is associated with better cross-validated models of neural activity. However, these simulations explore the effect of dimensionality in a highly idealized setting without attempting to capture the statistics of real DNNs or brains. Moreover, in our simulations, we explicitly controlled the type and level of noise in the representations, which makes the interpretation of ED straightforward, whereas, in real networks the potential contribution of noise is far more complex and there is no guarantee that ED will be related to the quality of latent dimensions. Nonetheless, these simulations can help us to generate testable hypotheses about the potential implications of

latent dimensionality in real networks, which we set out to explore next. Specifically, we sought to examine the relationship between latent dimensionality and encoding performance in state-of-the-art DNNs and recordings of image-evoked responses in visual cortex.

## Dimensionality in deep neural networks

Before presenting our analyses, it is helpful to first consider the interpretation of latent dimensionality in the context of DNNs and to highlight some important methodological details. The logic underlying dimensionality metrics like ED is that the scale of a dimension's variance is indicative of its meaningfulness, with high variance corresponding to meaningful dimensions and low variance corresponding to random or less-useful dimensions. Even in deterministic systems, like the DNNs examined here, variance scale can indicate meaningfulness. Indeed, previous work has shown that network training expands variance along dimensions that are useful for solving tasks while leaving unaltered, or even contracting, variance along random dimensions [21, 46–49]. This explains why networks trained with different random initializations end up with similar high-variance principal components but different low-variance components, and it explains why measures of network similarity appear to be more meaningful when they are weighted by variance [50, 51]. Furthermore, these empirical observations are consistent with theories of deep learning which argue that networks progressively learn the principal components of their training data and tasks, with the number of learned components increasing as a function of task and data complexity [48, 51–55].

It is important to note, however, that the scale of activation variance is not universally indicative of representation quality in DNNs. First, architecture-specific factors can affect ED in ways that are independent of learning, which means that ED values (and related dimensionality metrics) should only be compared across models with similar architectures [56]. Second, it is possible to arbitrarily rescale the variance of any dimension through normalization procedures (see S9 Text). Thus, to gain a better understanding of whether high-performing DNNs have expressive, high-dimensional representations or compressed, low-dimensional representations, it is important to examine the role of dimensionality while controlling for architecture. Here we controlled for architectural factors by focusing on a set of standard convolutional architectures (mostly ResNet). We provide a more detailed discussion of these points in S1 & S9 Text.

For our analyses, we examined a large bank of 568 layers from DNNs that varied in training task, training data, and depth. The training tasks for these DNNs included a variety of objectives, spanning both supervised (e.g., object classification) and self-supervised (e.g., contrastive learning) settings. We also included untrained DNNs. The training datasets used for these DNNs included ImageNet [57] and Taskonomy [58]. Most DNNs had ResNet50 or ResNet18 architectures [59]. We also examined a smaller set of models with AlexNet [60], VGG-16 [61], and SqueezeNet [62] architectures to ensure that our findings were not idiosyncratic to ResNets. We restricted our analyses to convolutional layers at varying depths because the structure of the fully connected layers substantially differed across models. A detailed description of all models is provided in S5 Text. Using this approach, we were able to examine the effect of ED while controlling for architecture.

We empirically estimated the ED of the DNNs by obtaining layer activations in response to 10,000 natural images from the ImageNet validation set [57]. We applied PCA to these layer activations and computed ED using the eigenvalues associated with the principal components. An important methodological detail is that we applied global average pooling to the convolutional feature maps before computing their ED. The reason for this is that we were primarily interested in the variance of image *features*, which indicates the diversity of image properties

that are encoded by each model, rather than the variance in those properties across space. Nevertheless, we show in S6 Text that our main results on the relationship between ED and encoding performance were observed even when ED was computed on the entire flattened feature maps without pooling (though to a lesser extent). The ED values that we computed can be interpreted as estimates of the number of meaningful dimensions of natural image representation that are encoded by each model (i.e., their latent dimensionality).

Our analyses of ED showed several general trends, which are discussed in detail in S8 Text. Briefly, we found that ED is higher for trained compared with untrained models, that ED tends to increase with layer depth, and that ED tends to be higher for models trained on a wide variety of natural images rather than only indoor scenes. These trends in ED suggest that feature expansion may be an important mechanism of the convolutional layers in DNNs.

## Dimensionality and encoding performance for neural data

We next wanted to determine if the ED of representational subspaces in DNNs was related to their encoding performance (Fig 2). To do so, we first compared DNN models with

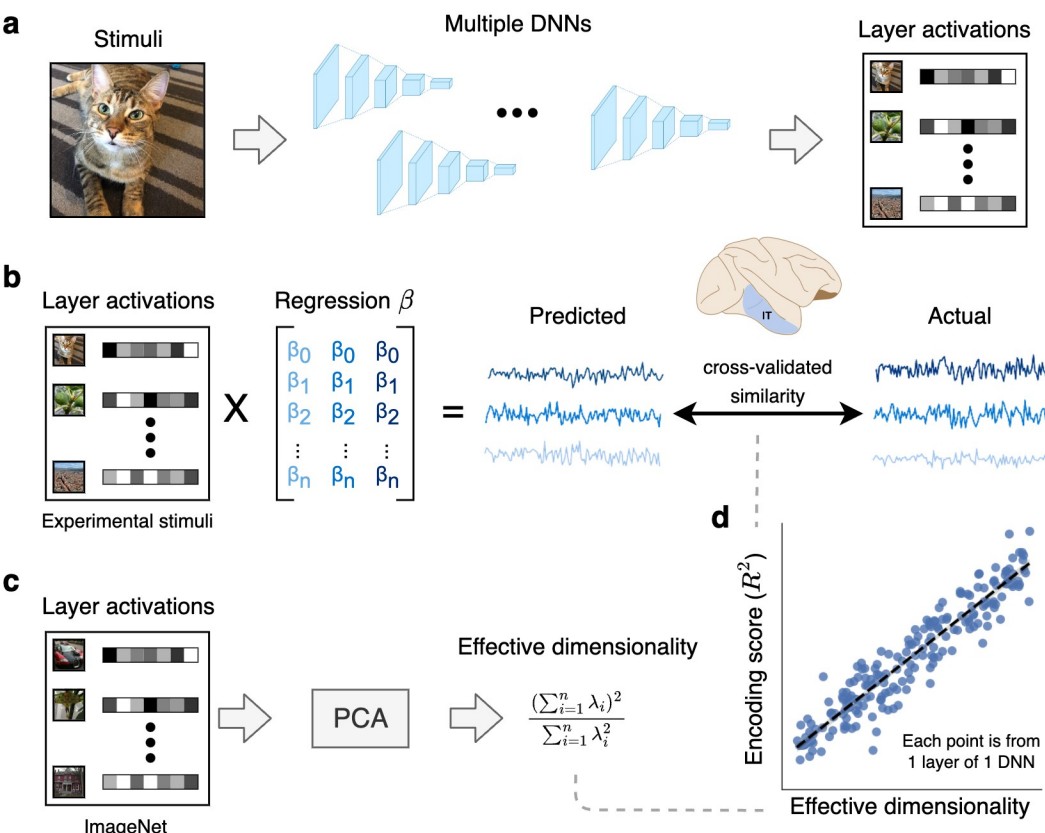

**Fig 2. Method for comparing latent dimensionality with encoding performance for neural data.** a: Layer activations were extracted from a large bank of DNNs trained with different tasks, datasets, and architectures. b: Using these layer activations as input, we fit linear encoding models to predict neural activity elicited by the same stimuli in both monkey and human visual cortex. We used cross-validation to evaluate encoding performance on unseen stimuli. c: To estimate the effective dimensionality of our models, we ran principal component analysis on layer activations obtained from a large dataset of naturalistic stimuli (specifically, 10,000 images from the ImageNet validation set). d: These analyses allowed us to examine the empirical relationship between effective dimensionality and linear encoding performance across a diverse set of DNNs and layers. DNN = deep neural network, PCA = principal component analysis. The brain icon in Fig 2 was obtained from https://doi.org/10.5281/zenodo.3926117.

electrophysiological recordings of image-evoked responses in macaque IT cortex—a high-level region in the ventral visual stream that supports object recognition [63]. These data were collected by [64], and the stimuli in this study were images of objects in various poses overlaid on natural image backgrounds. In total, the dataset consisted of 168 multiunit recordings for 3,200 stimuli. We quantified the ability of each convolutional layer in each DNN to explain neural responses by fitting unit-wise linear encoding models using partial least squares regression, which is a standard procedure in the field for mapping computational models to neural data [8]. These encoders were evaluated through cross-validation, where regression weights are learned from a training set and then used to predict neural responses to stimuli in a held-out test set (Fig 2b). We measured encoding performance by computing the median explained variance between the predicted and actual neural responses across all recorded units.

Our analysis revealed a clear and striking effect: the encoding performance of DNN models of high-level visual cortex is strongly and positively correlated with ED (Fig 3a). This effect was also observed when separately examining subsets of models from the same DNN layer (Fig 3b), which means that the relationship between ED and encoding performance cannot be reduced to an effect of depth. (Note that the reverse is also likely true: there may be effects of depth that cannot be reduced to effects of ED.) This within-layer analysis also perfectly

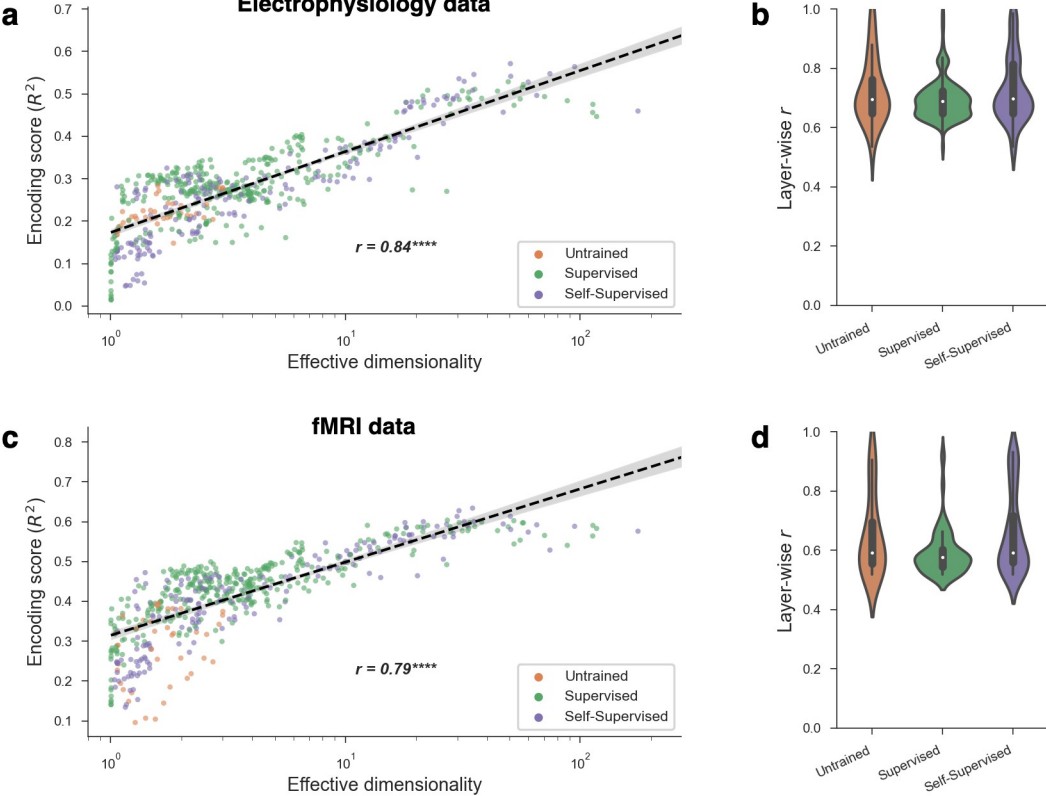

**Fig 3. Relationship between effective dimensionality and encoding performance.** a: The encoding performance achieved by a model scaled with its effective dimensionality. This trend also held *within* different kinds of model training paradigms (supervised, self-supervised, untrained). Each point in the plot was obtained from one layer from one DNN, resulting in a total of 568 models. Encoding performance is for the monkey IT brain region, which had the strongest relationship with ED among regions we considered. b: Even when conditioning on a particular DNN layer, controlling for both depth and ambient dimensionality (i.e., number of neurons), effective dimensionality and encoding performance continued to strongly correlate. The plot shows the distribution of these correlations (Pearson *r*) across all unique layers in our analyses. c,d: Similar results were obtained for human fMRI data.

controls for ambient dimensionality, which is the number of neurons in a layer, and, thus, shows that this effect is specifically driven by the *latent* dimensionality of the representational subspaces. Furthermore, this effect could not be explained by categorical differences across learning paradigms or across model training datasets because it was also observed when separately examining subsets of models that were either untrained, supervised, or self-supervised (Fig 3a and 3b) as well as subsets of models that were trained on Taskonomy or ImageNet (see S6 Text). Remarkably, this effect is not specific to the encoding model paradigm, as it was also observed when using representational similarity analysis [65], which involved no parameter fitting (see S7 Text). Finally, we also performed these analyses for human high-level visual cortex using an fMRI dataset collected by [66]. The human fMRI results for the lateral occipital region (LO; a high-level region supporting object recognition) are shown in Fig 3c and 3d, and they reveal a similar positive relationship between ED) and encoding performance in the human brain.

We further performed analyses in datasets for other lower-level visual regions in monkeys (V1, V2, and V4) and for multiple other visual regions in the human brain. While the relationship between ED and encoding performance was strongest in high-level visual cortex in both monkeys and humans, similar but weaker effects were also observed in multiple other visual regions, with the exception of monkey V1 (see S6 Text. We speculate that the effect was not observed in the monkey V1 dataset for two reasons. The first is that the stimuli in this V1 dataset were simple images of synthesized textures, which may not require the complexity of high ED models [67]. The second is that V1 is known to be explained by primitive edge detectors that likely emerge in most DNNs, even those with low ED. Another intriguing possibility is that each brain region could have an "optimal" value of ED where model encoding performance tends to peak (reminiscent of the "Joint regime" we showed in Fig 1c), and that this optimal value could increase along the cortical hierarchy [68]. Indeed, when we fit inverted-U shaped curves to the per-region data in S6 Text, the results suggest that the ED with peak encoding performance increases from V1 to V2 and from V4 to IT.

In addition to ED, we examined the complete eigenspectra of all models (i.e., the variance along successive principal components). Intuitively, models with more slowly decaying eigenspectra use more of their principal components to represent stimuli. In line with this, Fig 4a shows that the more slowly a model's eigenspectrum decays, the higher its encoding performance tends to be. Interestingly, many of the top-performing models tend to approach a power-law eigenspectrum decaying as $\frac{1}{i}$, where $i$ is the principal component index. This power-law decay corresponds to a proposed theoretical limit wherein representations are maximally expressive and high-dimensional while still varying smoothly as a function of changing stimuli [40].

While visual inspection of eigenspectra plots can be illuminating, it is difficult to succinctly summarize the large amount of information that these plots contain. We, therefore, continue to use ED in our discussions below because of the concise, high-level description it provides.

Finally, in S1 Text, we compared ED with other geometric properties, and we examined how all these properties are related to both encoding performance and object classification performance. Our findings show that ED is among the strongest geometric predictors of DNN performance metrics, suggesting that high-dimensional representational subspaces allow DNNs to perform a variety of tasks related to primate vision, including the prediction of image-evoked neural responses and the classification of natural images.

In sum, these findings show that when controlling for architecture, latent dimensionality is strongly linked to the encoding performance of DNN models of high-level visual cortex, suggesting that the richness of the learned feature spaces in DNNs is central to their success as computational models of biological vision.

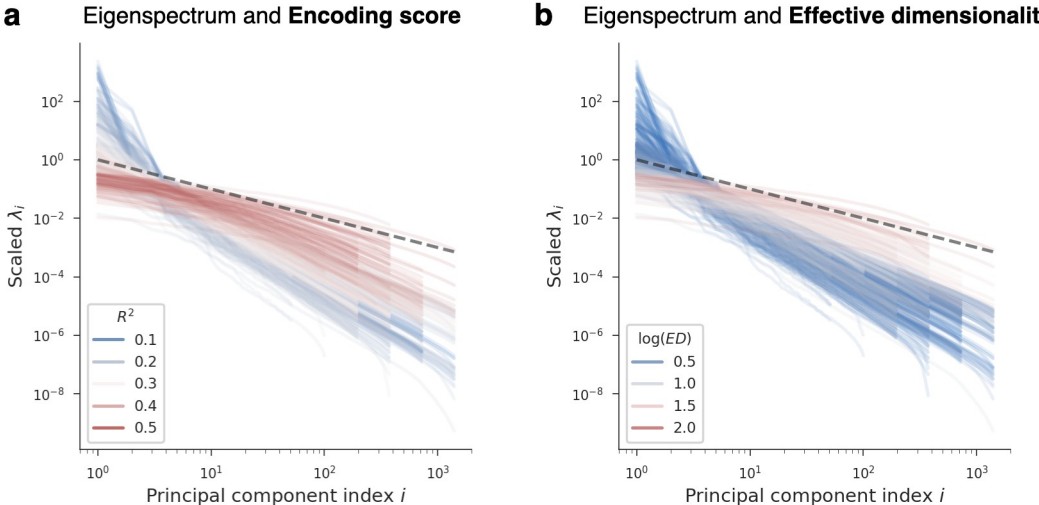

**Fig 4. Relationship between model eigenspectra and encoding performance.** Each curve shows the eigenspectrum of one layer from one DNN. The x-axis is the index of principal components, sorted in decreasing order of variance, and the y-axis is the variance along each principal component (scaled by a constant in order to align all curves for comparison). The black line is a reference for a power law function that decays as $\frac{1}{i}$, where $i$ is the principal component index. This power law of $\frac{1}{i}$ was hypothesized in [40] to be a theoretical upper limit on the latent dimensionality of smooth representations. a: Eigenspectra are color-coded as a function of the corresponding encoding performance each model achieved. Models with more slowly decaying eigenspectra (i.e., higher latent dimensionality) are better predictors of cortical activity, with top-performing models approaching the theoretical upper bound on dimensionality proposed in [40]. Encoding performance is for the IT brain region, which had the strongest relationship with ED among regions we considered. b: Eigenspectra are color-coded as a function of their corresponding ED. Since ED is a summary statistic of an eigenspectrum meant to quantify its rate of decay, models with more slowly decaying eigenspectra tend to have higher ED.

## High dimensionality is associated with better generalization to novel categories

Our finding that top-performing encoding models of high-level visual cortex tend to have high dimensionality was surprising given that previous work has either not considered latent dimensionality [2, 4–14] or argued for the opposite of what we discovered: namely that low-dimensional representations better account for biological vision and exhibit computational benefits in terms of robustness and categorization performance [20, 37]. We wondered whether there might be some important computational benefits of high-dimensional sub-spaces that have been largely missed in the previous literature. Recent theoretical work on the geometry of high-dimensional representations suggests some hypotheses [16, 40, 44]. Specifically, it has been proposed that increased latent dimensionality can improve the learning of novel categories, allowing a system to efficiently generalize to new categories using fewer examples [16]. Efficient learning is critical for visual systems that need to operate in a diversity of settings with stimulus categories that cannot be fully known *a priori*, and it is something that humans and other animals are remarkably good at [69–72].

Thus, we examined whether the dimensionality of our DNNs was related to their generalization performance on newly learned categories. We employed a standard transfer learning paradigm in which we fixed the representations of our DNN models and tested whether they could generalize to a new target task using only a simple downstream classifier (depicted in Fig 5a). Following the approach in [16], we trained a classifier on top of each model's representations using a simple prototype learning rule in which stimuli were predicted to belong to the nearest class centroid. We used 50 object categories from the ImageNet-21k dataset [73] and

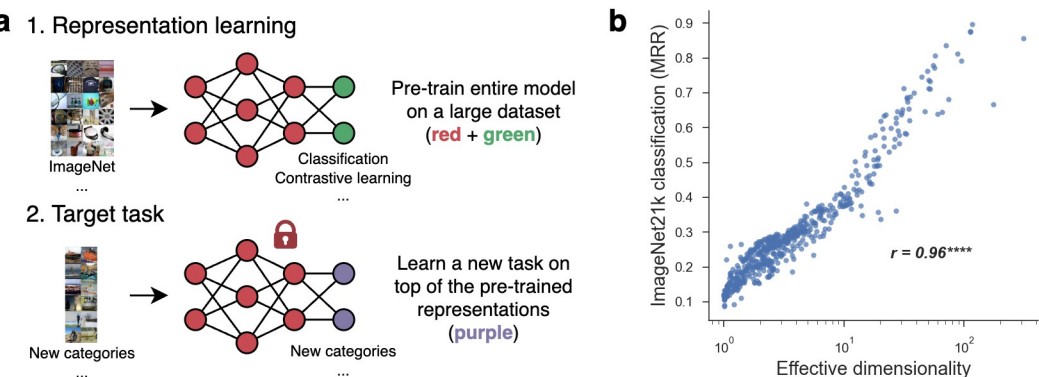

**Fig 5. The computational benefit of high effective dimensionality in generalization to new object categories.** We examined the hypothesis that high-dimensional representations are better at learning to classify new object categories [16]. a: We tested this theory using a transfer learning paradigm, where our pre-trained model representations were fixed and used to classify novel categories through a prototype learning rule. The images for the ImageNet tiles were cropped from the following open source: https://cs.stanford.edu/people/karpathy/cnnembed/. b: High-dimensional models achieved substantially better accuracy on this transfer task, as measured using the mean reciprocal rank (MRR).

trained on 50 images per category, evaluating on another 50 held-out images from the same categories. Importantly, none of the ImageNet-21k classes appeared in any of the datasets our models were pre-trained on, allowing us to assess a relationship between the latent dimensionality of a representation and its ability to classify novel categories. The results, illustrated in Fig 5b, show a striking benefit of high-dimensionality for this task. Even though high-dimensional representations have traditionally been thought to be undesirable for object classification [17, 21–25], they proved to be extremely effective in separating novel categories. This suggests that while low-dimensional representations may be optimal for performing specialized tasks (such as separating the fixed set of categories in the standard ImageNet training set), high-dimensional representations may be more flexible and better suited to support open-ended tasks [16, 43, 47, 74, 75].

## High dimensionality concentrates projection distances along linear readout dimensions

How do high-dimensional models achieve better classification performance for novel categories? If we consider an idealized scenario in which category instances are distributed uniformly within unit spheres, it is a geometric fact that projections of these subspaces onto linear readout dimensions will concentrate more around their subspace centroids as dimensionality increases [16, 76, 77]. The reason for this is that in high dimensions, most of the subspace's mass is concentrated along its equator, orthogonal to the linear readout dimension. This blessing of dimensionality is typically referred to as the *concentration of measure phenomenon* [76], and we depict it for the case of an idealized spherical subspace in Fig 6a and 6b (see Section Materials and methods).

Although we do not know the geometric shapes of category subspaces in DNNs or whether they can be approximated by elliptical subspaces, we can, nonetheless, test whether there is empirical evidence that a similar concentration phenomenon occurs in our models. To answer this question, we computed the average sample projection distance between every pair of our 50 ImageNet-21k classes, normalized by an estimate of the subspace radius for each class (see Section Materials and methods). This yielded a matrix of all pairwise projection distances for

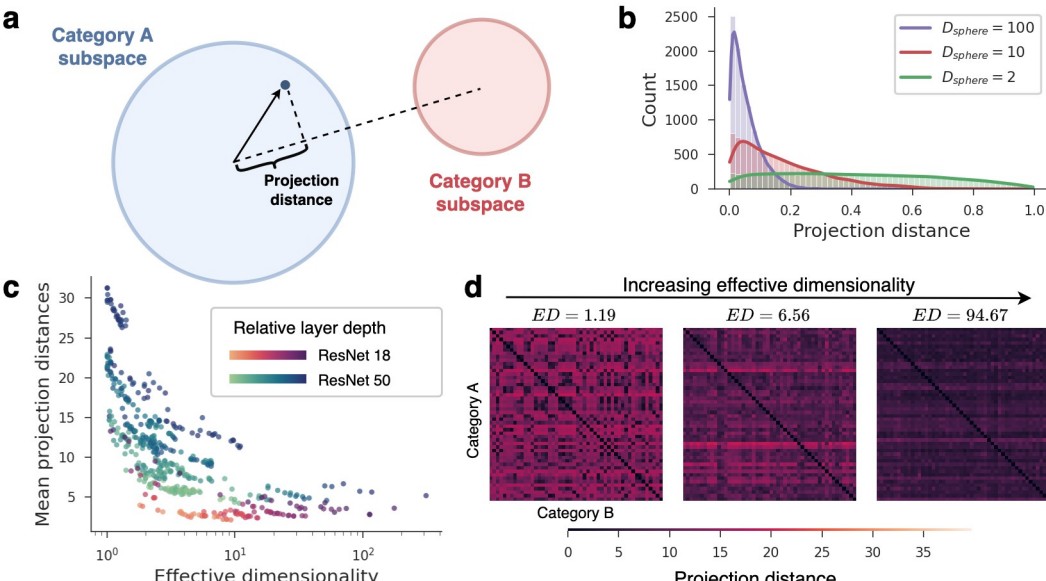

**Fig 6. High-dimensional models concentrate sample projections close to their class centroids.** a: For binary classification, the projection distance of a sample refers to the sample's distance from its class centroid along the classification readout direction, normalized by the subspace radius. b: For idealized spherical subspaces, the distribution of projection distances concentrates more tightly around 0 as the dimensionality $D_{sphere}$ increases. c: Empirically, the mean projection distances of our models decreased as effective dimensionality increased, matching what is predicted by theory. Note that because the magnitude of projections partially depend on the model architecture and layer depth (denoted by different colors), projection distances form distinct bands in the plot. However, when looking only at models with the same architecture and layer (i.e., looking at points sharing the same color), projection distances reliably decrease with ED. d: Full projection distance matrices, computed along classification readout directions between all object category pairs. Matrices are shown for three different models of increasing effective dimensionality.

each model. Fig 6c shows that, as predicted, the mean projection distance systematically decreases as model ED increases. This means that sample projection distances concentrate closer to class centroids as model dimensionality increases, allowing DNNs with higher ED to discriminate novel categories more effectively. This concentration effect exemplifies an under-appreciated computational advantage conferred by the geometric properties of high-dimensional subspaces.

## Discussion

By computing geometric descriptors of DNNs and performing large-scale model comparisons, we discovered a geometric phenomenon that has been overlooked in previous work: DNN models of high-level visual cortex benefit from high-dimensional latent representations. This finding runs counter to the view that both DNNs and neural systems benefit by compressing representations down to low-dimensional subspaces [20–39, 78]. Furthermore, our findings suggest that DNN models of high-level visual cortex are best understood in terms of the richness of their natural image representations rather than the details of their training tasks and training data.

Our results speak to a fundamental question about the dimensionality of neural population codes. Empirical estimates have consistently shown that the latent dimensionality of both DNNs and neural systems is orders of magnitude lower than their ambient dimensionality (i.e., the number of neurons they contain) [15, 20, 22, 29, 33–37, 78, 79]. Furthermore, there

are compelling theoretical arguments for the computational benefits of low-dimensional codes, which may promote robustness to noise [21, 24, 25, 27, 30, 31], abstraction and invariance [23, 28, 32, 38, 79], compactness [15], and learnability for downstream readouts [80]. It is, thus, no surprise that many neuroscientists and machine learning researchers have argued that a signature of optimal population codes is the degree to which they reduce latent dimensionality, focusing on the minimum components needed to carry out their functions. However, there are competing theoretical arguments for the benefits of high-dimensional population codes. High-dimensional codes are more efficient [42, 45], they allow for the expression of a wider variety of downstream readouts [43], and they may have counterintuitive benefits for learning new categories due to concentration phenomena in high dimensions [16]. Furthermore, recent evidence from large-scale data in mouse visual cortex suggests that cortical population codes are higher dimensional than previously reported and may, in fact, approach a theoretical limit, above which the code would no longer be smooth [40]. Our findings provide strong evidence for the benefits of high-dimensional population codes. Specifically, we demonstrated two major benefits that are directly relevant to computational neuroscientists. First, high-dimensional DNNs provide more accurate cross-validated predictions of cortical image representations. In fact, when looking at the eigenspectra of our top-performing models, they appear to approach the upper limit on dimensionality that was proposed in [40]. Second, high-dimensional DNNs are more effective at learning to classify new object categories. We suggest that while low-dimensional codes may be optimal for solving specific, constrained tasks on a limited set of categories, high-dimensional codes may function as general-purpose representations, allowing them to better support an open-ended set of downstream tasks and stimuli.

Our findings also have implications for how neuroscientists interpret the relationship between DNNs and neural representations. When developing theories to explain why some DNNs are better than others as computational brain models, it is common for researchers to place a strong emphasis on the task on which the network was optimized [2, 4–14]. We and others have found that a variety of tasks are sufficient to yield high-performing encoding models of visual cortex [10–12, 14]. However, when analyzing the geometry of these networks, we found that a common thread running through the best-performing models was their strong tendency to encode high-dimensional subspaces. It is worth emphasizing that we were only able to observe this phenomenon by analyzing the geometry of latent representations in many DNNs with the same architecture and examining large-scale trends across these models. In other words, the most important factors for explaining the performance of these models were not evident in their task-optimization constraints but rather in their latent statistics. This finding is consistent with other recent work showing that widely varying learning objectives and architectures—including transformer architectures from computational linguistics—are sufficient to produce state-of-the-art encoding performance in visual cortex, which suggests that task and architecture are not the primary explanation for the success of DNNs in visual neuroscience [11, 12, 14]. Our findings are also consistent with recent work that calls into question the apparent hierarchical correspondence between DNNs and visual cortex [81, 82]. Indeed, we found that the relationship between latent dimensionality and encoding performance generalized across layer depth, meaning that even within a single layer of a DNN hierarchy, encoding performance can widely vary as a function of latent dimensionality. Our work suggests that the geometry of latent representations offers a promising level of explanation for the performance of computational brain models.

A recent preprint examined the ED of layer activations across a highly varied set of neural networks and found that while ED was higher for trained versus untrained versions of the same architectures, as was also shown here, variation in ED across diverse trained networks

was not correlated with encoding performance in visual cortex [14]. However, we want to emphasize that this latter finding is not unexpected given the nature of the model set examined in the study. Specifically, the authors examined the best performing layer from models with diverse architectures and varied normalization layers, including convolutional networks, transformers, and multilayer perceptrons, all of which can have architecture-specific effects on the shape of representational eigenspectra (see Section Dimensionality in deep neural networks and S9 Text). Importantly, the existence of such architecture-specific effects means that a given ED value will not have the same meaning across all networks, and, for this reason, we caution against comparing ED values (and related dimensionality metrics) across networks with highly varied architectures [56]. Furthermore, the ED metric computed in Conwell at al. is different from the ED metric that we found to be most informative in our own analyses. Specifically, we computed ED based on channel covariance by using global pooling, whereas Conwell at al. computed ED based on a mixture of both channel and spatial covariance, which in our models predicts encoding performance less well (see S6 Text). We focused on channel covariance because it reflects the richness of feature types in a network and because it has been shown to be highly informative for theories of learning in convolutional architectures [48]. In contrast, spatial covariance reflects the spatial smoothness of layer activations, which is informative in its own right but appears to be less useful for understanding the richness of DNN representations. In sum, we argue that the most informative approach for understanding the role of dimensionality in DNNs is to examine how variations in dimensionality are related to variations in model performance while controlling for architecture. We further propose that channel covariance, rather than spatial covariance, offers the best lens on the key representational properties of DNNs.

We should also note that because we focused specifically on channel covariance in our main analyses, our notion of dimensionality is different from the dimensionality metrics used in most previous studies of CNNs, which generally did not make a distinction between channel and spatial information [15, 20, e.g.]. We believe that channel and spatial covariance have different functional consequences for visual representations, and it may be important to keep the distinction between these two factors in mind when examining dimensionality and when comparing results across studies. Nonetheless, we did not find evidence in our study to suggest that dimensionality trends in CNNs are markedly different when spatial information is included —-rather when including spatial information in our analyses, we found qualitatively similar, though weaker, relationships between dimensionality and measures of model performance. We suggest that a useful direction for future work is to better understand the specific ways in which channel and spatial covariance might differentially affect task performance in neural networks.

Our results raise several important considerations for future work. First, while our findings show that computational models of visual cortex benefit from high latent dimensionality, our method cannot speak to the dimensionality of visual cortex itself and was not designed to do so. Indeed, the theory that we presented in Section Dimensionality and alignment in computational brain models predicts that high-dimensional DNNs should generally better explain neural activity *even if neural representations in the brain are low-dimensional*. However, our findings suggest a promising model-guided approach for tackling this issue: one could use high-dimensional DNNs to create stimulus sets that vary along as many orthogonal dimensions as possible. This sets up a critical test of whether the latent dimensionality of visual cortex scales up to the dimensionality of the model or, instead, hits a lower-dimensional ceiling.

Second, we found that one way in which DNNs can achieve both strong encoding performance and strong image classification performance is by increasing the latent dimensionality of their representations. However, this finding diverges from previous work that has linked

better classification performance to dimensionality reduction in DNN representations [20, 21]. We believe that this discrepancy arises due to a fundamental problem with classification metrics: DNNs with the best classification scores are optimal for a single task on a small and closed set of categories (e.g., ImageNet classes), but these seemingly optimal DNNs may be less useful for representing new categories or for representing meaningful variance within a category (e.g., object pose). This problem with classification metrics may help to explain why the strong correlation between DNN classification performance and cortical encoding performance [8, 12] appears to break down at the highest levels of classification accuracy [83–85] (see an extended discussion of these issues in S1 Text). Future work should move beyond classification accuracy and explore richer descriptions of representational quality.

Third, it is important to keep in mind that ED can vary as a function of architecture-specific factors and that some classes of operations can break the relationship between ED and the diversity of learned features, as we show in S9 Text. Thus, high ED is not universally associated with better DNN performance and comparisons of ED across different architectures may not be informative. In our analyses, we were able to reveal the benefits of high latent dimensionality by controlling for architecture. An exciting direction for future work is the development of new metrics of representational richness whose interpretation is independent of architecture.

Finally, an open question is whether our results are specific to convolutional neural networks and higher visual cortex or whether similar results could be obtained for other classes of computational models (e.g., transformers) and other sensory and cognitive domains (e.g., audition, language). Note that our approach of using global pooling to focus on channel covariance means that our methods are tailored to convolutional architectures. Thus, further work will be needed to apply our approach to other classes of architectures in which channel and spatial information are not readily separable.

In sum, we propose that the computational problems of real-world vision demand high-dimensional representations that sacrifice the competing benefits of robust, low-dimensional codes. In line with this prediction, our findings reveal striking benefits for high dimensionality: both cortical encoding performance and novel category learning scale with the latent dimensionality of a network's natural image representations. We predict that these benefits extend further and that high-dimensional representations may be essential for handling the open-ended set of tasks that emerge over the course of an agent's lifetime [16, 43, 47, 74, 75].

## Materials and methods

### Simulations

The theory and rationale behind our simulations are explained in S2 Text. Precise implementation details are provided in S3 Text. Additional simulation results are provided in S4 Text.

### Deep neural networks

We used 46 different DNNs, each with either a ResNet18, ResNet50, AlexNet, VGG-16, or SqueezeNet architecture. Training tasks included supervised (e.g., object classification) and self-supervised (e.g., colorization) settings. We also used untrained models with randomly initialized weights. The training datasets of these DNNs included ImageNet [57] and Taskonomy [58]. Further details describing each model are provided in S5 Text. Convolutional layers in ResNets are arranged into 4 successive groups, each with a certain number of repeated computational units called blocks. We extracted activations from the outputs of each of these computational blocks, of which there are 8 in ResNet18 and 16 in ResNet50. For other architectures, we used layers that were close to evenly spaced across the depth of the model. Across our 46 DNNs, this resulted in a total of 568 convolutional layers that we used for all further analyses.

## Neural datasets

Neural responses were obtained from a publicly available dataset collected by [64]. Two fixating macaques were implanted with two arrays of electrodes in IT—a visual cortical region in later stages of the ventral-temporal stream—resulting in a total of 168 multiunit recordings. Stimuli consisted of artificially-generated gray-scale images composed from 64 cropped objects belonging to 8 categories, which were pasted atop natural scenes at various locations, orientations, and scales. In total, the dataset held responses to 3,200 unique stimuli.

In S6 Text, we also show results on additional datasets. The V4 electrophysiology dataset was collected in the same study as for IT [64]. The V1 electrophysiology dataset was collected by [67], and consisted of responses to 9000 simple synthetic texture stimuli. In addition to our electrophysiology datasets, we also used a human fMRI dataset collected by [66]. The stimulus set consisted of 810 objects from 81 different categories (10 object tokens per category). fMRI responses were measured while 4 subjects viewed these objects, shown alone on meaningless textured backgrounds, and performed a simple perceptual task of responding by button press whenever they saw a "warped" object. Warped objects were created through diffeomorphic warping of object stimuli [86]. The methods for identifying regions of interest in these data are detailed in [66]. The localizer scans for these data did not contain body images, and, thus, a contrast of faces-vs.-objects was used to select voxels from the parcel for the extrastriate body area (EBA).

## Predicting neural responses

We obtained activations at a particular layer of a DNN to the same stimuli that were used for obtaining neural responses. The output for each stimulus was a three-dimensional feature map of activations with shape *channels × height × width*, which we flattened into a vector. For our monkey electrophysiology dataset, we fit a linear encoding model to predict actual neural responses from the DNN layer features through partial-least-squares regression with 25 latent components, as in [8] and [83]. To measure the performance of these encoding models, we computed the Pearson correlation between the predicted and actual neural responses on held-out data using 10-fold cross validation, and averaged these correlations across folds. We aggregated the per-neuron correlations into a single value by taking the median, which we then normalized by the median noise ceiling (split-half reliability) across all neurons. This normalization was done by taking the squared quotient $r^2 = (r/r_{ceil})^2$, converting our final encoding score into a coefficient of explained variance relative to the noise ceiling. The entire process described above for fitting linear encoding models was implemented with the Brain-Score package [83, 84] using default arguments for the [64] public benchmark.

The process for fitting voxel-wise encoding models of human fMRI data (presented in S6 Text) differed slightly from the above. For each of our 4 subjects, we used 9-fold cross-validated ordinary least squares regression. Encoding performance was measured by taking the mean Pearson correlation between predicted and held-out voxel responses across folds, and then aggregated by taking the median across voxels. Finally, this score was averaged across subjects. No noise-ceiling normalization was used.

Before fitting these linear encoding models, we also applied PCA to the input features in order to keep the number of parameters in the encoders constant. For each model layer, principal components were estimated using 1000 images from the ImageNet validation set. Layer activations to stimuli from the neural dataset were then projected along 1000 components and finally used as regressors when fitting linear encoding models. We emphasize that this dimensionality reduction procedure was done purely for computational reasons, as using fewer regressors reduced the computer memory and time required to fit our encoding models.

Our findings are not sensitive to this particular decision, as we obtained similar results by applying average-pooling instead of PCA to our DNN feature maps as an alternative method for reducing the number of regressors.

### Estimating latent dimensionality

We used a simple linear metric called effective dimensionality (ED) [87–90] to estimate the latent dimensionality of our model representations. ED is given by a formula that quantifies roughly how many principal components contribute to the total variance of a representation. We, thus, ran PCA on the activations of a given model layer in response to a large number of natural images (10,000 from the ImageNet validation set) in an effort to accurately estimate its principal components and the variance they explain. An important methodological detail is that we applied global average-pooling to the convolutional feature maps before computing their ED. The reason for this is that we were primarily interested in the variance of image *features*, which indicates the diversity of image properties that are encoded by each model, rather than the variance in those properties across space.

### Classifying novel object categories

To see how model ED affected generalization to the task of classifying novel object categories, we used a transfer learning paradigm following [16]. For a given model layer, we obtained activations to images from $M = 50$ different categories each with $N_{train} = 50$ samples. We then computed $M$ category prototypes by taking the average activation pattern within each category. These prototypes were used to classify $N_{test} = 50$ novel stimuli from each category according to a simple rule, in which stimuli were predicted to belong to the nearest prototype as measured by Euclidean distance. This process was repeated for 10 iterations of Monte Carlo cross-validation, after which we computed the average test accuracy. Importantly, none of these object categories or their stimuli appeared in any of the models' pre-training datasets. Stimuli were taken from the ImageNet-21k dataset [73], and our object categories were a subset of those used in [16].

### Projection distances along readout dimensions

Section High dimensionality concentrates projection distances along linear readout dimensions investigated how points sampled from high-dimensional object subspaces project along linear readout vectors. First, we illustrated this phenomenon in Fig 6b using simulated data. We sampled $N$ points uniformly in the volume of a sphere of dimensionality $d$ and radius 1. For each point, we projected it along a random unit vector in $\mathbb{R}^d$, forming a distribution of projection distances from samples in the sphere to its centroid. We then plot this distribution for increasing values of $d$.

For the experimental results in Fig 6c and 6d, we sampled 50 random images from the same 50 ImageNet-21k object categories described earlier. For every pair of object categories $i$ and $j$, we created a classification readout vector $\mathbf{w}^{i,j}$ by taking the difference between category centroids normalized to unit length:

$$\mathbf{w}^{i,j} = \frac{\mu(X^i, \dim = 1) - \mu(X^j, \dim = 1)}{\|\mu(X^i, \dim = 1) - \mu(X^j, \dim = 1)\|^2},$$

where $X \in \mathbb{R}^{50 \times d}$ is a matrix of model activations for a particular object category and $\mu(\cdot, \dim = 1)$ computes the category centroid by taking the mean along the row dimension. This is the readout vector along which samples would be projected using a prototype learning

classification rule. We then projected each sample $k$ in category $i$ along the readout vector, yielding a scalar projection distance $p_k^{i,j}$ to the category centroid:

$$p_k^{i,j} = |(\mathbf{x}_k^i - \mu(X^i, \dim = 1)) \cdot \mathbf{w}^{i,j}|.$$

For each pair of categories $(i, j)$, we therefore had a vector of projection distances $\mathbf{p}^{i,j} \in \mathbb{R}^{50}$ coming from the 50 image samples in category $i$, and we took the average of this vector to give a mean projection distance. This mean projection distance was the summary statistic we were interested in, since theory predicts that it is both influenced by dimensionality and relevant to classification [16]. However, one issue is that we wished to compare this summary statistic across different architectures and layers, which might have different feature scales. To correct for this, we needed to normalize each model's mean projection distances by some normalizing factor that quantifies the feature scale. We computed this normalizing factor by taking the square root of the average variance for each category's representations, which conceptually can be thought of as quantifying category subspace's "radius" [16]. Specifically, for a given model, we computed:

$$R^i = \sqrt{\mu(var(X^i, \dim = 1), \dim = 2))},$$

where $var(\cdot, \dim = 1)$ computes the per-dimension variance. We refer to the resulting values as the mean normalized projection distances $\hat{p}^{i,j} = \frac{\mu(\mathbf{p}^{i,j})}{R^i}$. For $i, j = 1, .., 50$ object categories, this procedure yields a $50 \times 50$ matrix $\hat{P}$ for each model. Fig 6d shows examples of these matrices for models of increasing ED. The means of these matrices as a function of ED are shown for all models in Fig 6c.

## Supporting information

**S1 Text. Anticipated questions.** Q&A-style response to questions that we anticipate readers to have.
(PDF)

**S2 Text. Theory of latent dimensionality and encoding performance.** Detailed description of the theory of ED and encoding performance sketched out in Section Dimensionality and alignment in computational brain models.
(PDF)

**S3 Text. Implementation of simulations.** Details for how we generated the simulated data and fit encoding models in Section Dimensionality and alignment in computational brain models.
(PDF)

**S4 Text. Additional simulation results.** Results for additional analyses that we conducted by varying parameters of the simulations described in Section Dimensionality and alignment in computational brain models.
(PDF)

**S5 Text. Details of DNN models.** Descriptions of all DNN models used in our analyses.
(PDF)

**S6 Text. Additional analyses of ED and encoding performance.** Analyses that examine the relationship between ED and encoding performance under different conditions from those described in the main paper.
(PDF)

**S7 Text. ED and representational similarity analysis.** Comparing ED to neural data using representational similarity analysis instead of encoding performance.
(PDF)

**S8 Text. ED varies with model and training parameters.** Trends in ED across factors of variation in our models, such as layer depth and pre-training dataset.
(PDF)

**S9 Text. High ED alone is not sufficient to yield strong performance.** Nuanced discussion of the conditions under which our theory predicts that encoding performance will increase with model ED, along with supporting analyses.
(PDF)

**S10 Text. Other correlates of DNN encoding model performance.** Comparison of encoding performance to DNN features other than ED, such as classification performance and other geometric statistics of the representations.
(PDF)

## Author Contributions

**Conceptualization:** Eric Elmoznino, Michael F. Bonner.

**Data curation:** Eric Elmoznino.

**Formal analysis:** Eric Elmoznino.

**Funding acquisition:** Michael F. Bonner.

**Investigation:** Eric Elmoznino.

**Methodology:** Eric Elmoznino, Michael F. Bonner.

**Project administration:** Eric Elmoznino.

**Supervision:** Michael F. Bonner.

**Validation:** Eric Elmoznino.

**Visualization:** Eric Elmoznino.

**Writing – original draft:** Eric Elmoznino, Michael F. Bonner.

**Writing – review & editing:** Eric Elmoznino, Michael F. Bonner.

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
