## [Decision Letter · Decision Letter 0]

9 Sep 2023

Dear Mr. Elmoznino,

Thank you very much for submitting your manuscript "High-performing neural network models of visual cortex benefit from high latent dimensionality" for consideration at PLOS Computational Biology.

As with all papers reviewed by the journal, your manuscript was reviewed by members of the editorial board and by several independent reviewers. In light of the reviews (below this email), we would like to invite the resubmission of a significantly-revised version that takes into account the reviewers' comments.

We cannot make any decision about publication until we have seen the revised manuscript and your response to the reviewers' comments. Your revised manuscript is also likely to be sent to reviewers for further evaluation.

Sincerely,

Drew Linsley

Guest Editor

PLOS Computational Biology

Thomas Serre

Section Editor

PLOS Computational Biology

Reviewer's Responses to Questions

**Comments to the Authors:**

Reviewer #1: First, I’d like to applaud the authors on the beautiful paper. The results and presentation are super clear and I really like it.

### Summary

I’m a big fan of using geometric language to explain DNN related phenomena in mechanistic way. The results from this paper gave me further insights on how to interpret the previous neural regression results.

In this paper, the authors clearly pointed out a potentially neglected possibility that some models have better brain predictivity not because their representation is better aligned, but because their effective dimensionality is large.

The authors extensively benchmarked and compared DNN encoding models for neuronal responses in primate IT and human fMRI data from higher visual area. They found that the effective dimensionality of the DNN representation is a robust predictor of its encoding accuracy, across training tasks and datasets, even controlling for layer depth. They also found that the effective dimensionality strongly correlates with the ability of classifying novel categories per few-shot learning.

All in all, this paper provides computational evidence for the benefits of high-dimensional neural code. It also provides a new lens to interpret previous results of brain predictivity.

### Strength

- There are extensive controlled experiment for the main effect

1. Results in Fig3B F.1B showed that even when controlled for the training method and the layers, the higher-dimensional representations are better regressors for neurons. This answered my question of “is ED just another name for the depth of the layer”, since ED and layer depth are correlated.

2. The results from control experiments in I.1 and I.2 are super interesting, which showed that when trained on label-shuffled ImageNet the DNN acquired lower ED in its representation and lower predictive power of the brain.

3. Further, simply applying a linear transform to whiten the representation increases the Effective Dimensionality (ED) but doesn't enhance the predictive power. This demonstrates that the Effective Dimensionality may not be the ultimate answer; instead, it could be a proxy for something more profound and directly related to the richness of the representation.

- The power law decay of spectra and its relation to the theory in Stringer et al 2019 Nature is surprising and interesting.

- There are extensive citations and links to prior discussions of the dimensionality of the neural code. These references help contextualize the new evidence introduced by this work within the existing literature.

### Major Question

However, there is a major issue I had with the interpretation of the data.

Regarding figure F.2 I have some issue with the authors’ characterization “*With the exception of the monkey V1 data, we found a strong, positive relationship between ED and encoding performance across multiple datasets*”. and they gave their interpretation in L228-234.

This positive correlation is statistically true for V4 and IT, but when looking at the data, there is another interpretation which the authors didn’t mention in main text: the relation between ED and R2 looks like an inverse U shape, which is reminiscent of the middle panel of Figure 1.C

If I were the authors, I would not fit a linear line to the data in Figure F.2, instead I’ll fit a U shaped curve and inspect the peak of the U shape as the “optimal ED” for this dataset. This would better fit the trend of data for V1 and V4. Using this model, it seems to me that V1 has a smaller optimal ED than V4 and V4 smaller than IT. This will be consistent with two facts: 1) your observation in Fig. H.1 that the log of ED scaled near linearly with the depth of the layer in ResNet. 2) usually the earlier layer in DNN is a better predictor of V1 data and deeper layer better for IT, and the depth of the optimal layer for regressing each cortical area increase along the ventral stream.

Thus the alternative interpretation that can be drawn from the data is that each visual area has an optimal ED, where higher visual cortices are associated with a higher optimal ED. If we focus our attention on the higher visual area where the optimal ED is as high as or higher than what can be obtained from ResNet, we will only observe the rising phase of the inverse U shape. In other words, the encoding score positively correlates and increases with ED. If this interpretation is correct, it doesn't contradict your main conclusion; rather, it refines your picture and potentially makes it more nuanced and intriguing.

Independent lines of research argue that there is an increase in intrinsic tuning dimension along the ventral stream [1]. This has been characterized using online search approaches to explore neural tuning on a high-dimensional image manifold. The findings indicate that the tuning peaks for the inferotemporal cortex (IT) are sparser, sharper, and have more tuned dimensions than those of V1 (primary visual cortex). This progression in intrinsic tuning dimension, along with the progression of optimal ED of the CNN (Convolutional Neural Network) regressor, seems interconnected and worthy of exploration.

My intuition is that if the neurons are tuned for K feature dimensions and are invariant to the rest, then the CNN's latent dimensions need to have at least K effective feature dimensions to regress it. Consequently, the more tuned dimensions the neurons possess, the higher the dimension the CNN representation needs to have in order to predict it.

[1]: Wang B, Ponce CR. Tuning landscapes of the ventral stream. Cell Rep. 2022 Nov doi: 10.1016/j.celrep.2022.111595

In response, I would like to hear the authors' thoughts on the following points:

- In the revised version of the paper, it would be beneficial if the authors could directly address this alternative view and tailor some conclusions specifically to the higher visual cortex. For instance, the conclusion found in lines 309-311 might need to be revised to focus on the "higher visual cortex" rather than the entire ventral stream.

Another approach could involve providing additional validation through neural data.

- With regard to the V1 dataset, I recall the Freeman 2013 paper, which recorded texture responses from both V1 and V2. Is there an opportunity to perform the same evaluation for V2 neurons and incorporate the findings into Figure F.2?

- As for the fMRI dataset, could the plot for the V1 area be displayed in the same figure as Figure F.3? Including these additional data from earlier visual cortices will aid in either validating or refuting the question raised above.

Just want to mention that, even if the conclusion is specifically applicable to IT or higher visual cortices, the result and the paper remain highly intriguing. They provide a geometric perspective that allows us to examine prior findings in a new light.

### Minor Issue / Question

1. **Neural regression method with a spatial focus**

There are many methods for regressing neuronal responses to CNN activation tensors. As you mentioned, PLS is a good one, but not the only one. You have also showed the OLS result in F.4, which is consistent.

In some sense, the PLS and OLS get rid of the spatial structure of the tensor. Could you possibly provide some pilot data for neural regression using spatial factorized readout? Here are some reference using these kinds of spatial-channel factorized method (It’s understood that redoing the whole analysis maybe too costly)

****Neural system identification for large populations separating "what" and "where"**** https://arxiv.org/abs/1711.02653 (Factorized convolution)

****Factorized convolution models for interpreting neuron-guided images synthesis**** https://2022.ccneuro.org/view_paper.php?PaperNum=1034 (Factorized correlation / regression)

**Generalization in data-driven models of primary visual cortex** https://www.biorxiv.org/content/10.1101/2020.10.05.326256v2 (Gaussian readout)

These methods have been used widely in practise e.g. https://www.science.org/doi/10.1126/science.aav9436 and https://www.biorxiv.org/content/10.1101/2023.05.12.540591v1

2. Neural network architecture diversity.

The paper has conducted extensive experiments utilizing the ResNet architecture, known for its popularity and residual connections. Could it be possible or feasible to incorporate additional CNN architectures into the analysis, such as Inception, VGG, AlexNet, or CorNet? This inclusion might enhance our ability to interpret the Brain Score leaderboard, where a multitude of architectures are evaluated and compared. If we discover that effective dimensionality serves as a robust predictor for neural predictivity within the BrainScore framework, the findings could become even more intriguing and significant.

3. **Method details regarding projection distance.**

In L480-485, the authors described how they computed the normalized projected deviation. I’m still unsure if I can reproduce it by reading this text. specifically, I’m not sure what “mean standard deviation along all category subspace dimensions” refers to.

Maybe a few equations specifying how the normalization is done will be helpful.

Further I saw no method section corresponding to Figure 6b, a simple description of the experimental set up will be helpful.

4. In the main text, there's a statement concerning projection that I find unclear (lines 285-288). Specifically, the parenthetical comment “*(assuming that the radius is held constant)*” leaves me uncertain about what the term "radius" refers to in this context. In my understanding, the quantity that should be held constant is the summed or averaged variance across all dimensions. By distributing this variance over more dimensions, the variance on an individual projection dimension would be significantly reduced. Is my interpretation consistent with what the authors intended?

5. Regarding the organization of the paper, it might be beneficial to add a separate subsection dedicated to discussing Effective Dimensionality in the context of Deep Neural Network (DNN) representation. For example, a focus on the factors that influence ED would be quite interesting, and it would assist readers in evaluating the relationship between higher ED and increased predictivity. It would be valuable to explore in this section why using ED as a proxy for representation richness is effective, as well as what might undermine this approach. Lines 184-206 and 207-211 have already provided a concise version of this idea, but further elaboration could enrich the understanding of the topic.

6. On line 475, the reference to "Appendix 3" appears to be a typo or mislink, as there is no such appendix in the document. Please consider correcting this reference.

7. There seems to be a typo on line 468 involving "N_{test}". Please review and correct as needed.

Reviewer #2: Elmoznino and Bonner's study is a valuable contribution to the field, shedding light on the relationship between the geometry of Deep Neural Networks (DNNs) and their performance. They challenge existing notions by demonstrating that DNNs with higher-dimensional representations offer better explanations for cortical responses and enhanced adaptability to new training domains. The authors substantiate their claims using an impressive blend of multi-unit recordings from monkeys and BOLD-MRI data from humans, coupled with an extensive collection of resnets trained on both Imagenet and Indoor buildings datasets. The findings consistently reveal that increased effective dimensionality correlates with improved neural data explanation, particularly in monkey IT, though with varied strength across other areas such as V1, V4 and human BOLD-MRI.

The paper excels in introducing alignment pressure, effective dimensionality, and their potential relevance to encoding performance and feature learning. This makes it accessible even to readers less familiar with these concepts. Nevertheless, I have some substantial concerns and some minor points that warrant attention:

Major Concerns:

1) The paper predominantly highlights the pronounced pattern in monkey IT, while comparatively weaker effects are observed in V4 and human BOLD-MRI. Not showing at least some of this data in the main text is somewhat misleading.

2) The assertion that "network layers with a slower decay in eigenspectrum are better in explaining neural data" primarily holds for monkey IT, while weaker associations are seen in V1, V4, and human BOLD-MRI. Only presenting IT in the main text is misleading.

3) The authors propose that higher-tier areas, processing input from lower-tier areas, benefit most from higher dimensionality due to the robustly low-dimensional nature of the latter. This suggests a potential pattern wherein lower-tier areas exhibit lower dimensionality than higher-tier ones. Addressing this possibility and its potential implications vis-à-vis previous research would add depth to the discussion.

4) I miss a discussion of how results that show that the brain is low-dimensional could have been observed.

5) No V1 BOLD-MRI data is shown. This would be informative (maybe use an anatomical mask to get it if a field map is absent).

6) The authors use resnets because the convets are better comparable. That is reasonable but resnets have a very specific architecture. Would it be sensible to look at some other types of models with this as caveat? At least it would be good to observe that the pattern, higher ED results in higher encoding score is also present in a not skip-connection architecture.

Minor Remarks:

Page 3, Line 61: Reframing "computational models of visual cortex" to "models that explain neural data" offers greater precision in describing the authors' contributions.

Page 6: The caption of Figure 1 refers to a "Joint regime" containing two plots, yet these are not separately labeled within the caption.

Page 9, Line 215: Emphasize that although the analysis in 3b partly addresses differences in effective dimensionality across DNN layers, some distinctions may still arise due to inherent variations between layers.

Page 10: For Figure 3c, enhancing the color differentiation between the three conditions would aid clarity.

Page 11: Given the exceptional performance of the area illustrated in Figure 4, it's advisable to mitigate potential misinterpretation by providing context regarding its superiority.

Page 14, Line 317: Consider specifying that the downstream readout could pertain to V1.

Page 30: In Appendix C, abbreviations of formulas spanning four pages, with only "AP" defined in the main text, might benefit from being introduced closer to Figure C1 for improved coherence.

In conclusion, Elmoznino and Bonner's paper is a captivating exploration of DNN geometry's connection to performance, encapsulating both significant insights and areas necessitating further clarification.

Reviewer #3: This paper by Elmoznino and Bonner aims to explain the success of deep neural networks (DNNs) as computational models of vision. The standard explanation to this end has been “task performance” which is typically some kind of engineering benchmark like top X% accuracy on Imagenet. Here the authors suggest an alternative approach – dimensionality, which is basically the sparseness of the eigen-spectrum. This understanding could potentially lead to the development of better models of vision which would be impactful. I commend the authors on making their code etc open access which makes it much easier to review the study and is a contribution to the field. This is a high standard that others should aspire to. But there also are several issues, including a lack of replication of the reported effects (Conwell 2023), that dampen the impact of the authors’ proposal (see below).

The core claim of this study is that neural networks with high dimensionality are better computational models of vision. My review will be focused almost entirely the authors’ defense of this central claim.

Major issues:

1. Issues with the dimensionality metric for DNNs. There are at least 3 issues around the use of this specific metric (effective dimensionality, ED).

a. ED estimates the sparseness of the eigen spectrum of the covariance matrix of the data. Essentially this is an analytical way to remove the low magnitude eigen values (noise) from the high magnitude ones (signal). (Relatedy this is a very old metric that pre-dates citation 32. Please cite correctly). This metric makes sense when estimating dimensionality from noisy sources, like neural or fMRI data. It is unclear why the authors would choose to apply the same metric for a noiseless data source like a neural network.

b. The second concern is the specific way in which the ED metric was applied to DNN activations. Here the authors first apply a global pool to the convolutional maps from the DNNs before computing ED. This specific experimenter choice can completely change the overall inference about the dimensionality in the study. As an example, the authors discuss the untangling hypothesis across successive stages of the hierarchy (Cox, DiCarlo and others). They further and use the results of this study to argue that dimensionality in fact increases or gets more entangled (and not decreases, or less untangled) along the hierarchy. This comparison is not warranted because those previous studies were motivated by increase in invariance (mostly size, position and viewpoint which are encoded in spatial parameters) which the pooled version would not capture.

c. The third related concern about the choice of global pooling before estimating ED is that restricts the analysis to convolutional models only. As such this specific test would not be possible in older model classes (like hmax) or newer, SOTA models like transformer. This significantly limits the scope of this method to conv nets alone.

2. Issues with the central claim of the study. The authors here consider every layer of every base model (architecture + training regime) to obtain the key figure supporting the central claim of the study (Figure 3). As such every layer is considered as a “model” of the visual system. This is a very specific interpretation of a model of visual cortex and is otherwise highly unusual because most others would consider the best layer for a given base model as a putative model.

a. Similar correlation plots for best-layer only across the model set (for untrained and trained models separately)

b. Similar correlation plots for the best-layer only, without the global pooling

c. Generalization beyond Taxonomy models. Taxonomy models are in general bad models of the high-level vision (see Conwell 2023 arxiv, Murty 2022 Nature Comm)

d. A discussion on how this study relates with the recent Conwell (2023 arxiv) paper that did not find any relation between model encoding performance and latent dimensionality (Figure 5 and associated sections). It is important to reconcile a direct failure to replicate the stated claims of this report.

e. A comparison of effective dimensionality as an explanation of better model performance versus other competing hypotheses (for eg. engineering benchmarks like those on brain-score). The authors lay out the other explanations in the discussion (lines 338 onward) but do little to actual test these competing ideas directly.

Overclaims. The paper tends to overclaim the results in several places which must be addressed. For instance, the high positive correlation as such is only observed for high-level visual cortical regions. Low level regions (like V1) often show no relation with effective dimensionality. One can observe the pareto-ceilings even for V4 which has much lower correlation (probably drives by untrained models?. My strong advice is to tone down the claims of the study.

**Have the authors made all data and (if applicable) computational code underlying the findings in their manuscript fully available?**

Reviewer #1: Yes

Reviewer #2: Yes

Reviewer #3: Yes

PLOS authors have the option to publish the peer review history of their article (what does this mean?). If published, this will include your full peer review and any attached files.

Reviewer #1: No

Reviewer #2: **Yes: **H S Scholte

Reviewer #3: No
---

## [Decision Letter · Decision Letter 1]

30 Dec 2023

Dear Mr. Elmoznino,

We are pleased to inform you that your manuscript 'High-performing neural network models of visual cortex benefit from high latent dimensionality' has been provisionally accepted for publication in PLOS Computational Biology.

Best regards,

Drew Linsley

Guest Editor

PLOS Computational Biology

Thomas Serre

Section Editor

PLOS Computational Biology

Reviewer's Responses to Questions

**Comments to the Authors:**

Reviewer #1: I appreciate the substantial effort made to address the major concerns raised about the interpretation of the results. I’m pretty happy about accepting the paper as it is. Some highlights

- I like the change of limiting the interpretation of Effective Dimensionality to within the same architectural class. Comparison of ED across architectures and with the Conwell preprint will need more future works to dissect.

- The additional analyses in F.3 (quadratic fit, and inclusion of V2 data); F.6, (results without global pooling) and J.1 (comparison with other geometric metrics), all made the results of the paper much stronger.

One comment on the limitation of the analysis to convolutional framework (L447-452)

- I feel your analysis could be readily applied to vision transformer architecture, where the ‘token” dimension is the “spatial dimension” and the “embedding” dimension is the “channel” dimension. Then the spatial average pooling and analysis could be done to them in the same fashion.

There are some minor textual changes that I encourage the authors to make.

Minor edits

- L29 - “*neural network models of visual cortex”* may be better phrased as “existing neural network models ……” or “current neural network models ……”

I suggest this change because I think the conclusion of the paper applies to the existing CNN models, but it could be a limitation of all the current models i.e. none of the current models really align with the brain. Potentially future neural network models of neural data could live in the othere regime.

- On L247 & L251 & L261, the author name preceding the citation may need parentheses around them. Currently the format looks a bit strange.

Reviewer #2: Happy to read that the authors have now also included LO in the main story and explicitly analysed visual cortex, and have shifted the focus the story on high-level visual cortex.

The authors have incorporated all my comments and I have no further comments. Loved the paper, discussed the archive version extensively in the lab meeting.

Reviewer #3: In my previous review, I raised 3 big concerns with the core claims in the paper. The authors have now done sufficient work to respond to each of these issues. I do think that the core paper is now more rigorous, clarifies some of the specific user-defined choices in this study, and also differentiates itself better from other similar claims in the field.

At this point I am happy to recommend the paper for publication. I congratulate the authors on this work and hope to see more detailed (possibly quasi-causal) exploration of geometric properties and model performance.

**Have the authors made all data and (if applicable) computational code underlying the findings in their manuscript fully available?**

Reviewer #1: Yes

Reviewer #2: Yes

Reviewer #3: Yes

PLOS authors have the option to publish the peer review history of their article (what does this mean?). If published, this will include your full peer review and any attached files.

Reviewer #1: **Yes: **Binxu Wang

Reviewer #2: **Yes: **H S Scholte

Reviewer #3: No

---

## [Editor Report · Acceptance letter]

5 Jan 2024

PCOMPBIOL-D-23-01150R1 

High-performing neural network models of visual cortex benefit from high latent dimensionality

Dear Dr Elmoznino,

I am pleased to inform you that your manuscript has been formally accepted for publication in PLOS Computational Biology. Your manuscript is now with our production department and you will be notified of the publication date in due course.

With kind regards,

Anita Estes
